# PSPS: A Step toward Tamper Resistance against Physical Computer Intrusion

**DOI:** 10.3390/s22051882

**Published:** 2022-02-28

**Authors:** Qi-Xian Huang, Ming-Chang Lu, Min-Yi Chiu, Yuan-Chia Tsai, Hung-Min Sun

**Affiliations:** 1Institute of Information Systems and Applications, National Tsing Hua University, Hsinchu 30013, Taiwan; xiangg800906three@gapp.nthu.edu.tw (Q.-X.H.); s106065803@m106.nthu.edu.tw (M.-Y.C.); 2Department of Computer Science, National Tsing Hua University, Hsinchu 30013, Taiwan; johnlu@gapp.nthu.edu.tw; 3Institute of Information Security, National Tsing Hua University, Hsinchu 30013, Taiwan; chuckmariotsai@gmail.com

**Keywords:** cyber-attack, physical security system, computer intrusion

## Abstract

Cyberattacks are increasing in both number and severity for private, corporate, and governmental bodies. To prevent these attacks, many intrusion detection systems and intrusion prevention systems provide computer security by monitoring network packets and auditing system records. However, most of these systems only monitor network packets rather than the computer itself, so physical intrusion is also an important security issue. Furthermore, with the rapid progress of the Internet of Things (IoT) technology, security problems of IoT devices are also increasing. Many IoT devices can be disassembled for decompilation, resulting in the theft of sensitive data. To prevent this, physical intrusion detection systems of the IoT should be considered. We here propose a physical security system that can protect data from unauthorized access when the computer chassis is opened or tampered with. Sensor switches monitor the chassis status at all times and upload event logs to a cloud server for remote monitoring. If the system finds that the computer has an abnormal condition, it takes protective measures and notifies the administrator. This system can be extended to IoT devices to protect their data from theft.

## 1. Introduction

By 2019, according to PurpleSec [1], there were over 812 million malware programs worldwide, as shown in Figure 1. Many computers are affected by malware each day, and an increasing number of intrusion detection systems (IDS) target malicious activity of networks. For example, Security Onion [2] provides free tools that can monitor network packages and check network security, while Suricata [3] designs network-based intrusion detection systems (NIDS) that provide real-time intrusion detection.

### 1.1. Problem Statement

IDS, such as Snort [4] and OSSEC [5], typically monitor network traffic for suspicious or abnormal activity and send alerts to administrators. However, IDS should not only protect the network but also the machine itself, and IDS can provide better defense if physical secure protection is included. With Internet of Things (IoT) devices, attackers can use physical attacks to find new IoT vulnerabilities. Attackers initially buy a targeted IoT device and try to obtain physical access to it. They can then create a false attack to analyze the outputs through reverse engineering [6]. After finding vulnerabilities, attackers can use buffer overflow attacks against IoT devices to expose vulnerabilities of the system. For example, attackers can disassemble an IoT device and read-out the flash memory to find vulnerabilities in the software, and then tamper with the microcontroller to identify sensitive data or cause unexpected behavior. These actions allow them to gain complete control of IoT devices. The best defense is not only network intrusion detection, but a good response should include physical intrusion detection.

### 1.2. Research Contribution

We here simulate physical intrusion detection by a computer with sensor switches, which has future application to IoT devices. We design a physically secure protection system (PSPS) that protects sensitive data and gives an alert when someone attempts to open the chassis or tamper with the computer. This can help prevent data from being stolen or reverse engineered. We describe the PSPS methodology in Section 4.

At run-time, PSPS starts with a computer with sensor switches broken by an attacker, which is classified into physical intrusion types. PSPS then extracts the intrusion behavior received from sensor switches and shows an alert message on the website and sends an email and text message to the administrator. Our prototype PSPS sends an alert message to the administrator directly, or the alert message returned by PSPS can be used as input to traditional IDS to handle different types of intrusion (Figure 2).

### 1.3. Organization

The balance of this work is organized as follows. Section 2 presents the background knowledge of computer security, IDS, and physical security. Section 3 introduces related research on PSPS. Section 4 describes the proposed approach and system architecture, and Section 5 presents its implementation in detail. Chapter 6 uses test cases to verify PSPS, and conclusions are summarized in Section 7.

## 2. Background

This section first introduces the basic concept of computer security in Section 2.1. Section 2.2 describes vulnerabilities, Section 2.3 explains the contents of computer protection, and Section 2.4 focuses on hardware protection mechanisms. Section 2.5, Section 2.6, Section 2.7, Section 2.8 explain intrusion detection systems, including network-based intrusion detection systems and host-based intrusion detection systems. Section 2.9 introduces the concept of physical security. This background knowledge helps to better understand the technology of the proposed system.

### 2.1. Computer Security

Computer security [7] is a critical part of information security. Its main purpose is to protect computer data security, avoid data theft, and ensure that authority is granted only to legally authorized users.

### 2.2. Vulnerabilities

Vulnerabilities are defects in computer system security that threaten the confidentiality, integrity, availability, and access control of the system or its application data. Most vulnerabilities can be found in the common vulnerabilities and exposures (CVE) database [8]. Vulnerabilities can typically be classified into denial-of-service attack, backdoor, eavesdropping, privilege escalation, direct-access attacks, polymorphic attacks, phishing, social engineering, spoofing, and tampering. Table 1 shows the five most frequent computer security vulnerabilities in 2019 [9].

### 2.3. Computer Protection

In computer security, a security policy is to reduce possible harm as much as possible by eliminating or preventing threats, or to generate security reports to reduce threats [10]. Countermeasure types include hardware protection, security by design, security architecture, security measures, and vulnerability management. This section focuses on hardware protection mechanisms.

### 2.4. Hardware Protection Mechanisms

Most research focuses only on network security, but hardware is also a notable security issue. It may be considered more secure to use devices and methods such as computer chassis intrusion detection, dongles, trusted platform modules (TPM), USB ports disabled, mobile-enabled access, and drive locks. The main focus of this section is computer chassis intrusion detection.

### 2.5. Intrusion Detection System

An intrusion detection system (IDS) is software that can monitor systems or a network for malicious activity or policy violations [11]. If an IDS detects any intrusion activity, it reports this to administrators and takes protective measures. IDS can be classified into two types of systems:(1)Network intrusion detection systems (NIDS);(2)Host-based intrusion detection systems (HIDS),

These two can identify the attempts to break the security objectives, such as confidentiality, integrity, availability, and nonrepudiation [12]. We focus on the HIDS in a section that follows.

### 2.6. Network-Based Intrusion Detection System

An NIDS detects network intrusion activity through a system using data mining. Figure 3 shows the process of analyzing real network traffic data using such a system [13].

### 2.7. Host-Based Intrusion Detection System

HIDS usually only observe and audit system log files under the Windows NT environment when the monitoring system logs in a UNIX environment for malicious behavior. The abilities of HIDS follow:(1)Confirms whether the hacker has successfully intruded;(2)Monitors the activity of a specific host system;(3)Detects malicious activity in real time;(4)Does not require additional hardware equipment.

The drawbacks of HIDS are:(1)Low flexibility: All hosts may be installed with different operating platforms, and each operating system has different audit log files, so various HIDS must be installed for different hosts.(2)Restricted monitoring and scanning: HIDS can only see the network packet information received through the host.(3)Additional system resources: System resources of the monitored host are consumed during monitoring, which restricts performance of the host itself.

The main purpose of HIDS is to add security for vulnerable software. In contrast, the main purpose of IDS is to detect intrusion activity and send suspicious records to administrators as alarms. Figure 4 shows the HIDS process [14].

### 2.8. Intrusion Prevention System

An intrusion prevention system (IPS) is a computer network security device that monitors network packages or network devices. It can adjust, interrupt, or isolate some abnormal network packages in real time. IPS can be described as an extension of IDS since it not only detects suspicious activity, but also takes protective measures. Figure 5 shows the architecture of IPS [15].

### 2.9. Physical Security

The primary objects of physical security consideration are various hardware devices, equipment room environment, and other material carriers that can also be understood as hardware security. Hardware facilities are the basic conditions for carrying and implementing information system functions, so physical security is also the most direct and primitive attack target. If the server hardware has already fallen into the hands of the attacker, even a strict firewall policy and highly complex operation passwords are made useless. Any access to hardware can lead to security risks and become targets of attacks, for example, if the computer chassis is not closed, backup power is lacking, BIOS has not set a password, or there are disasters such as earthquake, fire, or flood.

## 3. Related Work

This section describes an existing IDS in Section 3.1 and an IPS design in Section 3.2 to explain our research motivation and design principles of the system.

### 3.1. Snort

Snort is a network intrusion detection and prevention system (NIDS and NIPS) written in C language. It was created in 1998 by Marty Roesch and is now a well-known open-source program. The state-of-the-art in IDS research is represented in the Institute of Electrical and Electronics Engineers (IEEE) Study on Network Intrusion Detection System (Snort) [16], which obtains a unified output for a series of network packets. Its architecture is shown below (Figure 6).

The current addresses an aspect of intrusion detection that is not a direct focus of the network packets. We also consider physical attacks where the computer chassis has been tampered with.

### 3.2. OSSEC

OSSEC is an open-source HIDS [17] that monitors important corporate servers and various applications to prevent attacks, abuses, and misuses of corporate resources. It has log analysis, file integrity check, policy monitoring, rootkit detection, real-time alarm, linkage response, etc. In addition, it is used in security event management (SEM) and security information management (SIM) solutions. The proposed system is also host-based, but we focus on the chassis status. We store the system logs in the database and upload them to a cloud server.

### 3.3. Intrusion Prevention System Design

We focus on the prevention of the intrusion by eliminating all sensitive data when the chassis is opened. Preventing servers from intrusion by malicious network packets has long been part of intrusion prevention systems (IPS) research. Zhang et al. (2004) describe the design of an IPS-based on SNMP [18].

In general, an IPS prevents the server from malicious intrusion by using blacklists to their IP. In contrast, we show how to protect sensitive data when the chassis has been tampered with and to prevent intrusion.

## 4. Methodology

This section describes the architecture of the proposed design. Section 4.1, presents the goal of the system. Section 4.2 introduces the system’s state diagram and the details of each state.

### 4.1. Goal of the System

Our goal is to propose a physically secure protection system that helps companies to prevent their sensitive data on the server from being stolen by breaking the computer or opening the chassis. Our system is to prevent outsiders or people who are not familiar with the system from stealing sensitive data. To acquire the convenience and lower the operating threshold, we build a website to monitor server status and control the system state.

### 4.2. State Diagram

There are four states in this system: initial, operating, maintenance, and locked. The five functions to change the four states are: initialize, maintain, lock, unlock, and protect. The state diagram is shown in Figure 7. Details of each state and function are explained in the following sections, and these functions are implemented in the Section 5.

#### 4.2.1. Initial State

The initial state is first in the system, so there are neither data nor programs to monitor and protect the server. The administrator needs to check the system and input the password on the website to initialize the system, starting the operating state, as shown in Figure 8.

#### 4.2.2. Operating State

The operating state is second in the system, where the system protects the server and sensitive data. If the server is tampered with or the chassis is opened, the system monitor detects it and locks the system (Figure 9), and sensitive data are removed. The monitor continues protecting the server until the problem is resolved. This is the main function in the system to prevent unauthorized data access.

During the operating state, if the administrator wants to upgrade programs or pause the monitor, the administrator inputs the password on the website, changing the system into the maintenance state (Figure 10). Without this step, the system may consider it to be an intrusion and change to the locked state.

#### 4.2.3. Locked State

In the locked state, the monitor continues protecting the server. All sensitive data or programs are removed. The administrator checks the server status to fix the problem, and then inputs the password on the website to unlock the system. After unlocking the system (Figure 11), the system state reverts to the maintenance state.

#### 4.2.4. Maintenance State

In the maintenance state, the system is paused. The administrator can upgrade programs or maintain the server in this state. After maintenance, the administrator should input the password on the website to restart the system and protect the server (Figure 12).

## 5. Implementation

This section shows the implementation of the system in detail. Section 5.1 lists the development environment that is prepared first. Section 5.2 describes the important basic functions in the system. Section 5.3, Section 5.4, Section 5.5, Section 5.6, Section 5.7, Section 5.8 explain the six modules in the system. Functions mentioned in the previous chapter are all developed from combinations of these six modules.

### 5.1. Development Environment

We develop this system with six modules: PSPS IntrusionManager, PSPS Defense, PSPS Synchronize, PSPS RecoveryManager, SPS IDSWeb, and PSPS IDSServer. We use the following three programming languages:(1)Visual C++ 2019;(2)Java 1.8.0;(3)HTML, CSS, JavaScript.

To develop the system, we use the following compilers and software:(1)Microsoft Visual Studio Community 2019;(2)Eclipse IDE for Enterprise Java Developers 2020-03;(3)MySQL Server 8.0;(4)Connector ODBC 8.0;(5)Apache Tomcat 8.5;(6)Hei-diSQL 11.0;(7)FileZilla Server;(8)SyncBack.

The experimental machine is an Advantech DPX-S1435(19) running on Windows 10 Enterprise with the following hardware settings: Intel i7-4770TE 2.3 GHz CPU, 8 GB RAM, and Intel(R) HD Graphics 4600) (Figure 13).

### 5.2. Important Basic Functions

There are eight intrusion line inputs in the machine, and we use three sensor switches plugged into three of intrusion line inputs, including Sensor 0, Sensor 6, and Sensor 7 (Figure 14). We can use the Advantech API to detect if any of the sensor switches are opened, which is the most important function of the system. The detection function of these sensor switches is shown in Figure 15.

Each sensor switch has a turn-on status and turn-off status, yielding eight status combinations (Figure 16). Sensor status 0 indicates all sensor switches are opened, and sensor status 193 indicates all sensor switches are closed. Our system considers only sensor status 193 to be safe, with other status combinations considering the machine to be unsafe.

If the machine is shut down, the sensor switch status is recorded in the EEPROM [20]. This can store some memory without electricity, so the EEPROM memory can be read one time after the machine is turned on. The EEPROM function is shown in Figure 17. The EEPROM function outputs are the same as the sensor status above.

### 5.3. PSPS IntrusionManager

PSPS IntrusionManager is responsible for detecting the sensor status. If it finds that a sensor status is unsafe, it calls the PSPS Defense to protect the sensitive data and also sends an alert message and email to the administrator. PSPS IntrusionManager first calls the EEPROM function to check the sensor status in the EEPROM, and then calls the sensor function to detect the sensor status every second. It saves all the sensor status in the local database and calls PSPS Synchronize. The administrator can modify the seconds, frequency, file location, and other parameters through the INI file.

SMSto API [21] sends the SMS message and JavaMail API [22] sends the email. They are packaged into two runnable jar files for more convenient use. SMSto API is shown below (Figure 18).

### 5.4. PSPS Defense

PSPS Defense is responsible for stopping programs, clearing the database, and removing files to prevent the sensitive data from being accessed. We design an API so the program can shut down normally to avoid data corruption due to forced termination. The administrator can modify the program location, folder location, database, and other parameters through the INI file.

### 5.5. PSPS Synchronize

PSPS Synchronize uploads the sensor status and the system state to PSPS IDSServer and clears uploaded data in the local database in order to remotely monitor the status of the server. The administrator can modify the seconds, frequency, database, and other parameters through the INI file.

Important data are backed up to another computer through SyncBackFree [23], which is a set of free and powerful file synchronization backup software, which synchronizes the backup of new and modified files, backs up files to other hard disks, folders, and even FTP servers, and can also synchronize two folders. It supports Unicode, and its functions are quite extensive. It is free without restrictions and can be used for commercial purposes.

### 5.6. PSPS RecoveryManager

PSPS RecoveryManager changes the system state when instructions and the password are input. For example, if the server is in the initial state, the initialize instruction and the password can be input to change the system state from initial state to operation state.

### 5.7. PSPS IDSWeb

PSPS IDSWeb shows the sensor status and system state on the website, and the password can be input to call PSPS RecoveryManager and change the system state (Figure 19). This website only shows in localhost in order to monitor the system easily.

### 5.8. PSPS IDSServer

PSPS IDSServer is a cloud server. It can receive the sensor status and system state uploaded from the local database to monitor the system state from any location.

When the server is operating normally, the system state is as shown in Figure 20. PSPS IntrusionManager is the most important module in the system since it continues detecting the sensor status at all times.

When PSPS finds a suspected intrusion, it triggers PSPS Defense to protect the data (Figure 21).

## 6. Experimental Results

After defining the purpose of the experiment, this section evaluates the proposed system security using two test cases. Results of the test cases are then discussed.

### 6.1. Purpose

The main purpose is to determine whether this system can discover that the computer’s chassis has been opened, and then clear all the sensitive data and programs. Two test cases are used to evaluate the system.

### 6.2. Test Case 1

Here, an unauthorized person opened the chassis when the server was running (Figure 22). The IntrusionManager module detected the intruded behavior and called the defense module to protect the server by stopping the server and clearing sensitive data. The proposed system protected the server in this case.

### 6.3. Test Case 2

An unauthorized person first shut down the computer and then opened the chassis. When the server restarted, the IntrusionManager module detected the intrusion with the EEPROM, then stopped the server and cleared sensitive data (Figure 23). Although in this case the system could not detect the intrusion in time, it kept the records after the computer restarted and changed it to the locked state.

### 6.4. BitLocker Protection

Another situation was considered in which the chassis was opened and the intruder removed the hard disk drive to access sensitive data on the hard disk drive. For this, BitLocker protected the disk by providing encryption for entire volumes, so the intruder could read only data in the test machine.

### 6.5. Unlock Process

In these test cases, the system enters the locked state. To unlock the system, the administrator must first fix the intruded status of the server, and then input the password in PSPS IDSWeb. After the system confirms that the server is safe, it can switch into the normal operating state (Figure 24).

### 6.6. PSPS IDSWeb Process

We next show the system state in the PSPS IDSWeb. We can monitor each sensor switch status (Figure 25). PSPS IDSWeb can not only monitor the sensor status but also control the system state. It connects with PSPS RecoveryManager to change the system state.

When the system is not running, PSPS IDSWeb is in the initial state, waiting for the administrator to initialize the system (Figure 26). First, it checks if the sensor switch status is safe, and clears the old record in EEPROM. If all of the system’s settings are finished, the system state can be initialized.

After the system is initialized by the administrator, the system switches to the operating state (Figure 27), and PSPS IntrusionManager begins to protect the server.

When the system detects an intrusion, it switches to the locked state, while the IntrusionManager module is still protecting the server (Figure 28).

After the administrator unlocks the system or wants to upgrade the programs, the system enters the maintenance state, at which time the IntrusionManager module stops protecting the server (Figure 29).

### 6.7. Intrusion Alert

When PSPS IntrusionManager detects the intrusion behavior, it notifies the administrator via email and SMS. We use Java Email API and SMSto to provide these two functions. We package them into two runnable jar files, so that we can easily use these functions.

### 6.8. Comparison

The proposed PSPS system is compared with two well-known IDS systems: HIDS-OSSEC and NIDS-SNORT, as shown in Table 2.

Table 2 simply shows the differences between different IDS, not their advantages and disadvantages; the most appropriate system depends on the functions required.

## 7. Conclusions

In this work, we propose a security system to protect against physical computer intrusion, in contrast to previous papers that focus on network intrusion. We use Advantech DPX-S1435 to perform several experiments. Through the three sensor switches, this system can detect most of the behaviors used to enter the computer and has effective corresponding measures to prevent the theft of sensitive data. We also present two test cases to verify the system operation and show how to monitor the system state with PSPS IDSWeb. Additionally, we provide a comparison with two well-known IDS systems: HIDS-OSSEC and NIDS-SNORT. Finally, we include two more sections for deeper discussion and future work.

### 7.1. Discussion

PSPS prevents unauthorized persons from accessing system data. However, insiders or persons familiar with the system architecture can disassemble the EEPROM, sensors, and hard disk to avoid PSPS detection. Overcoming this is an important future work.

### 7.2. Future Work

In the future, the system can be enhanced as follows:(1)Build a website for PSPS IDSServer to monitor several systems;(2)Encrypt the Ini file to enhance security;(3)Use a smart card to encrypt and decrypt the password in PSPS RecoveryManager;(4)Apply this system to IoT devices;(5)Provide more protection for the EEPROM and sensor switches.

## Figures and Tables

**Figure 1 sensors-22-01882-f001:**
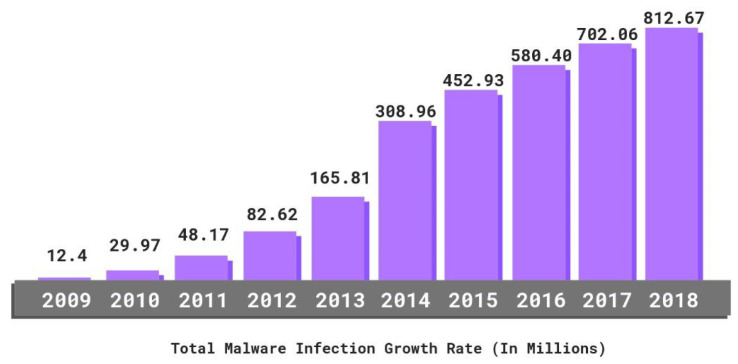
Total malware inflection growth rate [1]: many computers are affected daily by malware, and an increasing number of intrusion detection systems (IDS) target network malicious activity.

**Figure 2 sensors-22-01882-f002:**
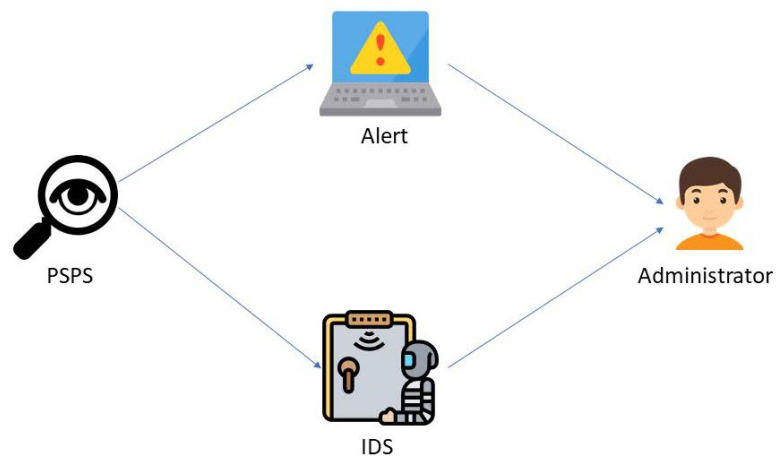
PSPS prototype: PSPS sends an alert message to the administrator directly. Alternatively, the alert message is returned by PSPS. It can be used as an input to traditional IDS in order to handle different types of intrusion.

**Figure 3 sensors-22-01882-f003:**
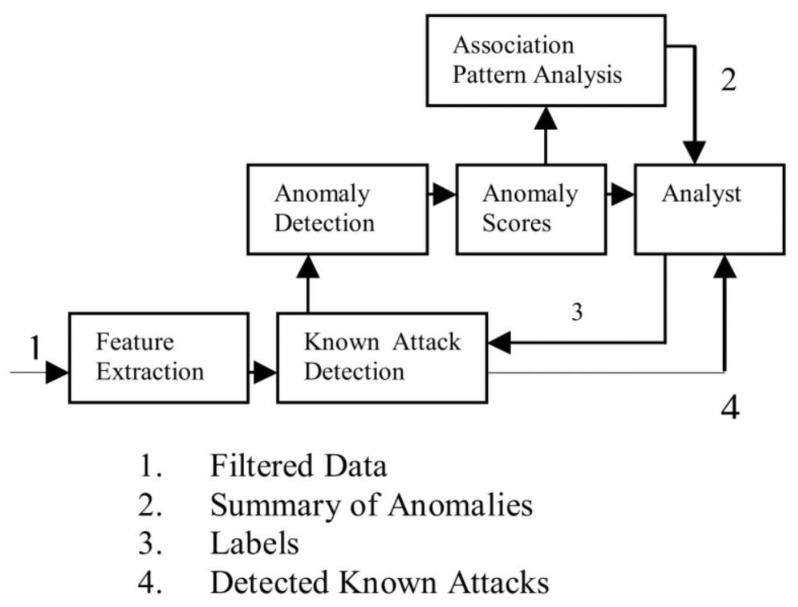
NIDS process flowchart: (1) filtered data to be the input data and extract the feature patterns, (2) check the association anomaly pattern analysis, (3) mark labels after analyst, and (4) detect known attack results.

**Figure 4 sensors-22-01882-f004:**
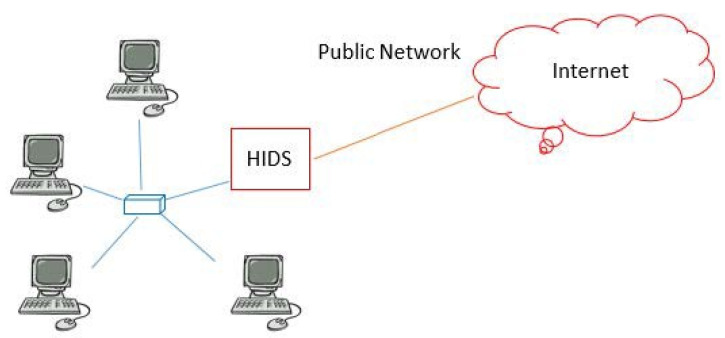
HIDS process: HIDS helps some vulnerable software add a security mechanism. The main purpose of IDS is to detect intrusion activity and send suspicious records to administrators by way of an alarm.

**Figure 5 sensors-22-01882-f005:**
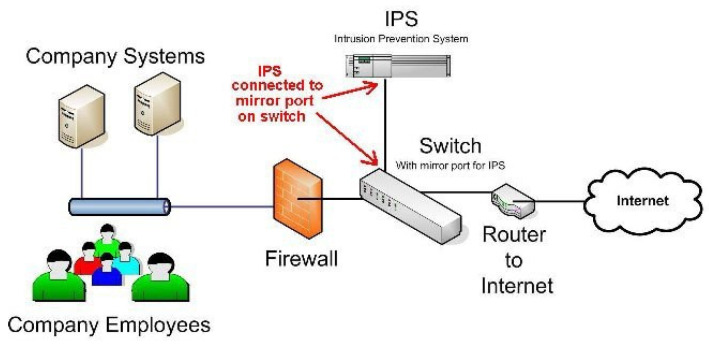
IPS architecture: IPS can be considered as an extension of IDS; it not only detects suspicious activity, but also takes protective measures.

**Figure 6 sensors-22-01882-f006:**
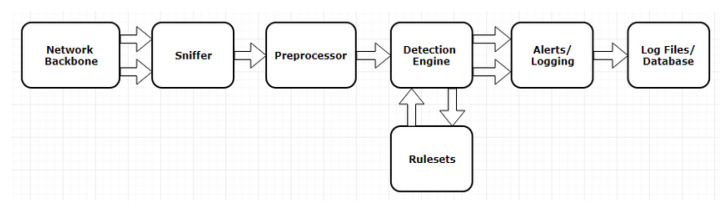
Snort architecture: Snort is a packet sniffer. However, it is designed to take packets and process them through the preprocessors. Each packet observed on the network is first passed through a set of preprocessors, which may extract information and/or modify the packet, and then check those packets against a series of rules (through the detection engine). Then, detection plugins match the packet against signature conditions. If a match was found, the information is sent through the alert system.

**Figure 7 sensors-22-01882-f007:**
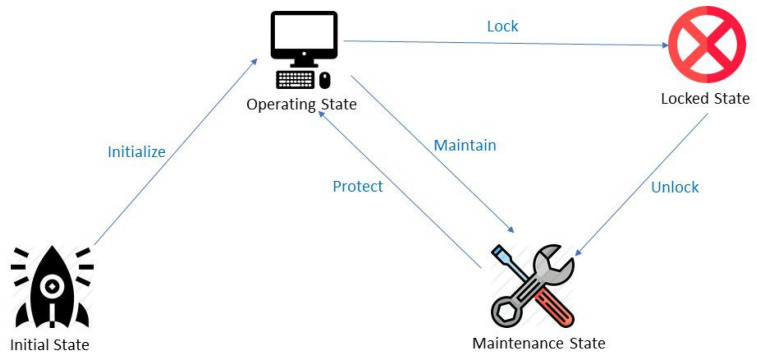
State diagram: There are four states in our system, including initial state, operating state, maintenance state, and locked state, and there are also five functions to change the four states, including initialize function, maintain function, lock function, unlock function, and protect function.

**Figure 8 sensors-22-01882-f008:**
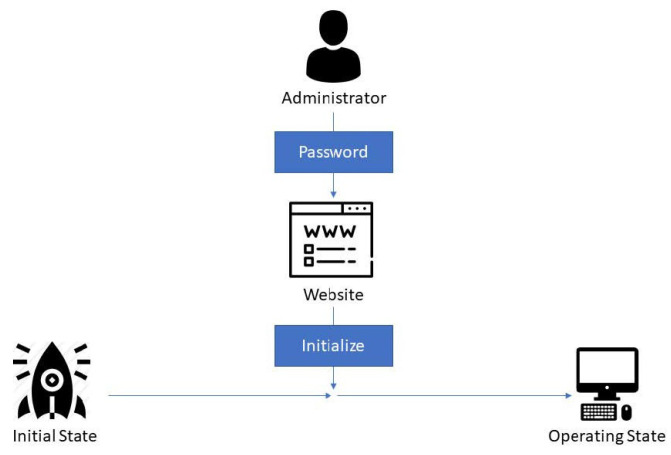
Initialize function: Initial state is the first state in the system. There are neither data nor programs to monitor and protect the server.

**Figure 9 sensors-22-01882-f009:**
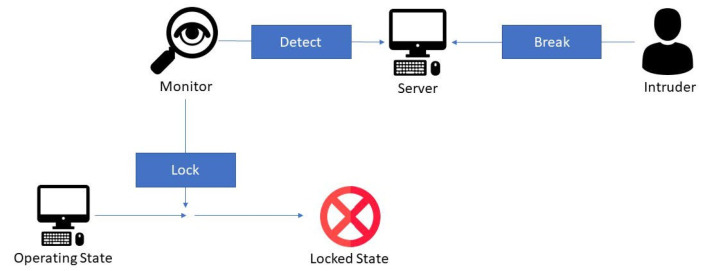
Diagram of the lock function process.

**Figure 10 sensors-22-01882-f010:**
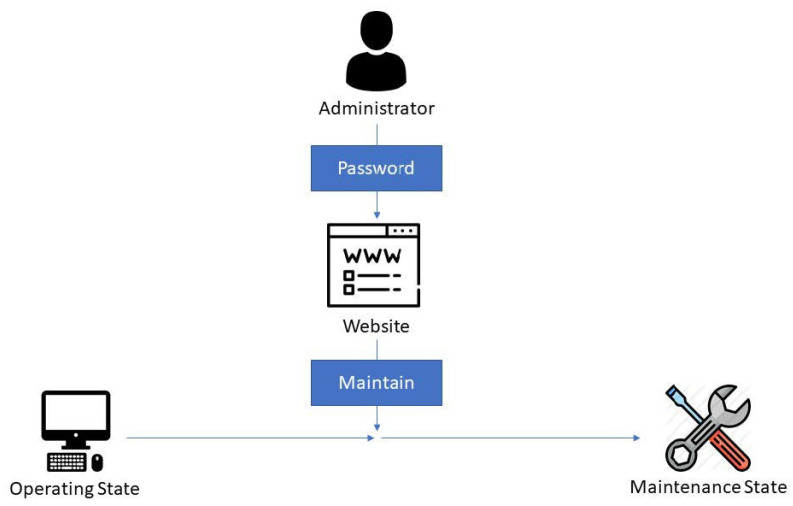
Maintain function: If the administrator wants to upgrade the programs or pause the monitor, the administrator inputs the password on the website and changes the system state into the maintenance state.

**Figure 11 sensors-22-01882-f011:**
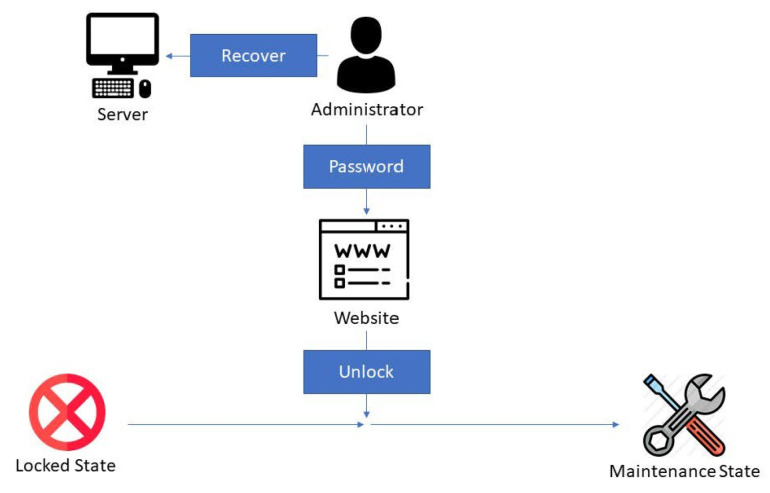
Unlock function: In the locked state, the monitor keeps protecting the server. All the sensitive data or programs are removed. The administrator should check the server status and fix the problem, and then input the password on the website to unlock the system.

**Figure 12 sensors-22-01882-f012:**
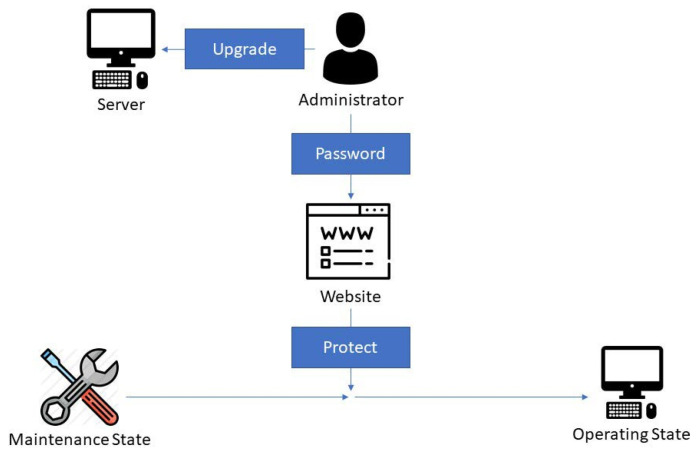
Diagram of the maintenance state for the protect function process.

**Figure 13 sensors-22-01882-f013:**
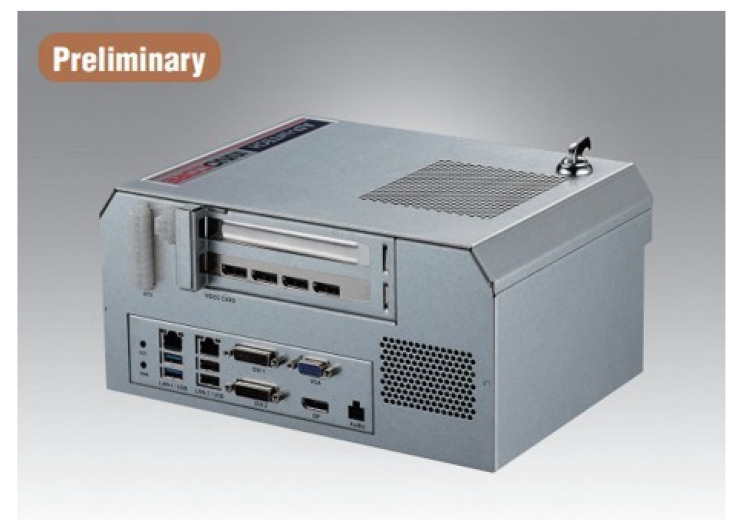
Advantech DPX-S1435 product (photograph courtesy of Advantech) [19].

**Figure 14 sensors-22-01882-f014:**
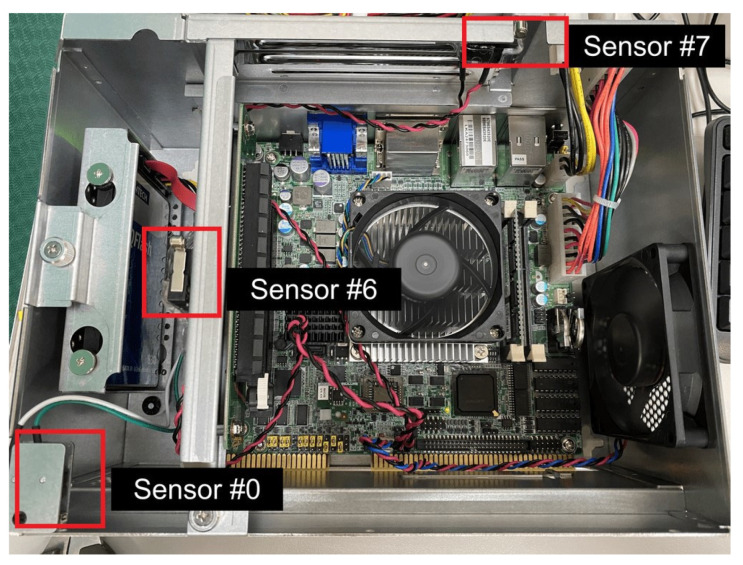
Advantech sensor switches [19].

**Figure 15 sensors-22-01882-f015:**
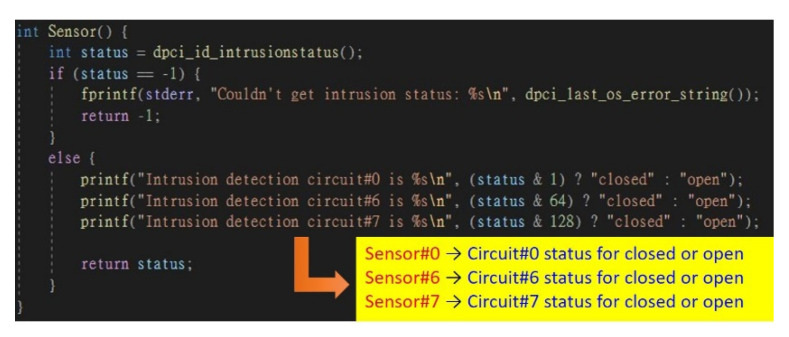
Sensor detection function.

**Figure 16 sensors-22-01882-f016:**
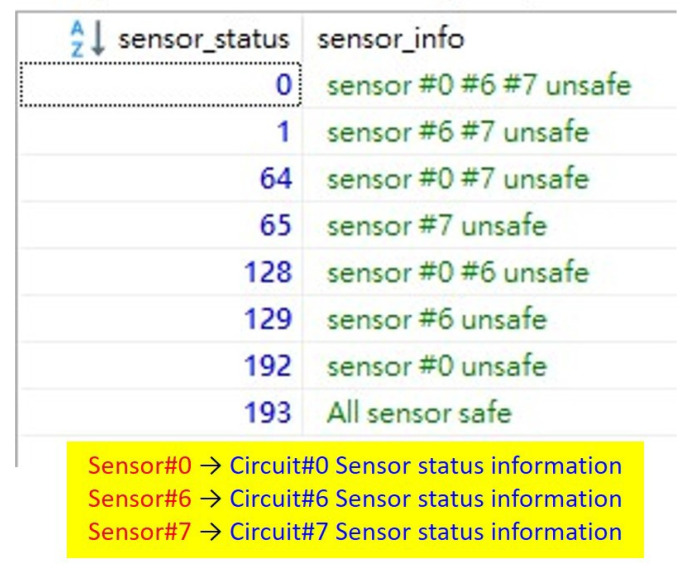
Eight sensor status.

**Figure 17 sensors-22-01882-f017:**
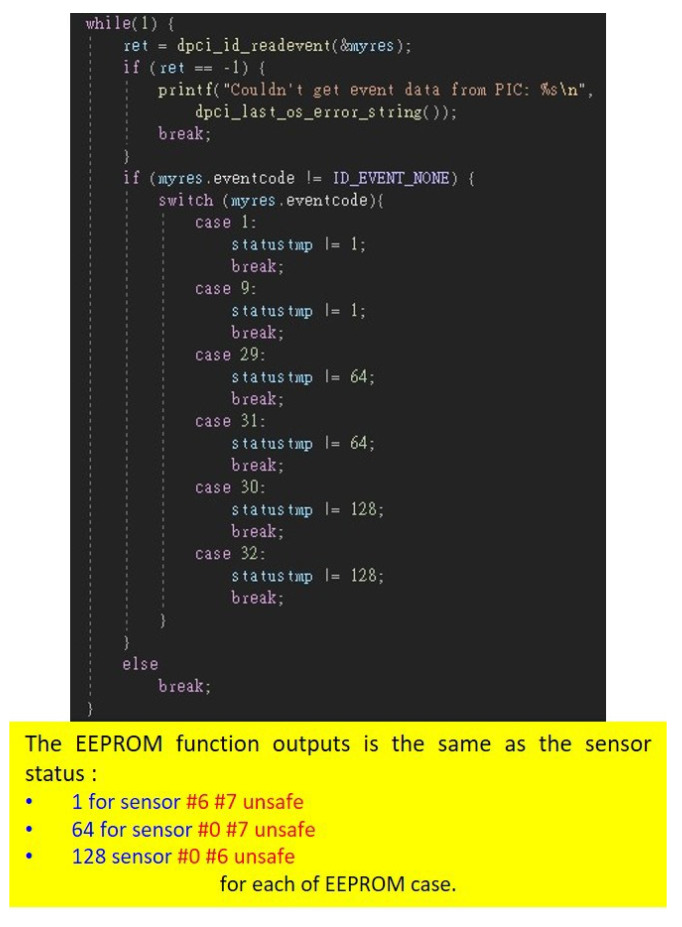
EEPROM function.

**Figure 18 sensors-22-01882-f018:**
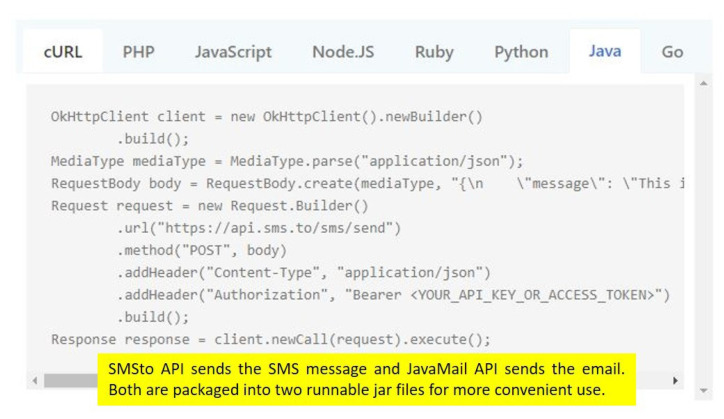
SMSto API interface.

**Figure 19 sensors-22-01882-f019:**
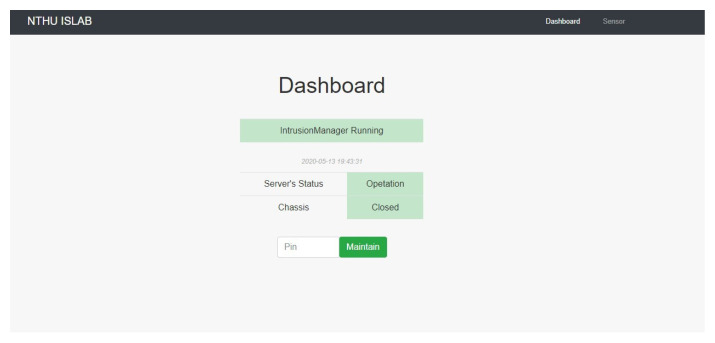
PSPS IDSWeb.

**Figure 20 sensors-22-01882-f020:**
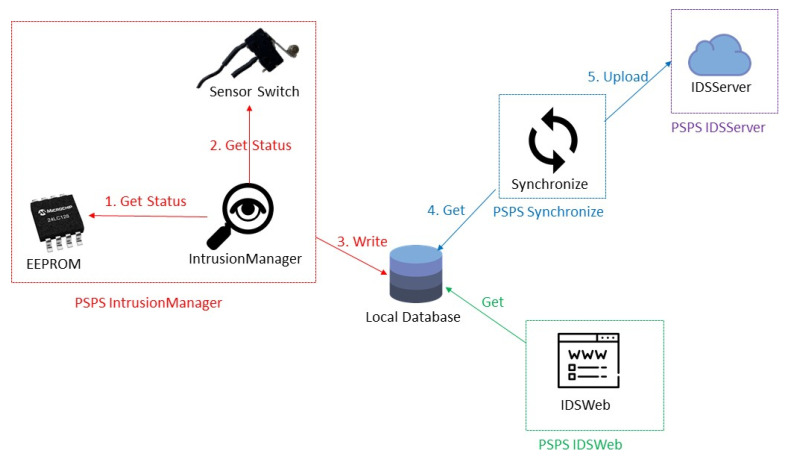
System operating: PSPS IDSServer is a cloud server. It can receive the sensor status and the system state uploaded from the local database. PSPS IntrusionManager is the most important module in the system; it constantly detects the sensor status.

**Figure 21 sensors-22-01882-f021:**
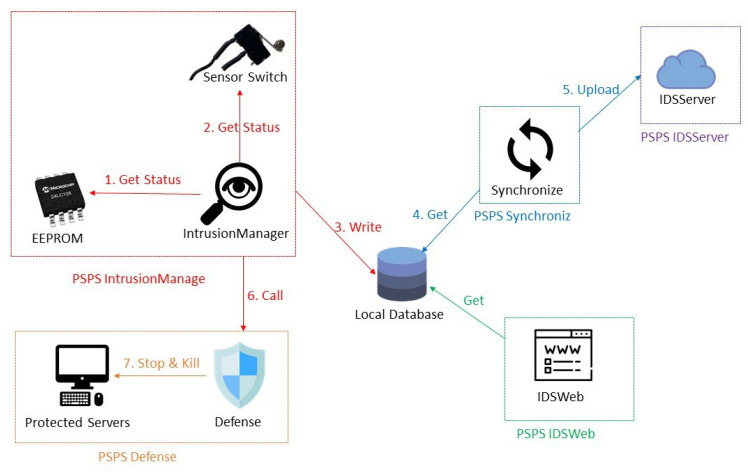
Diagram of the system locking process.

**Figure 22 sensors-22-01882-f022:**
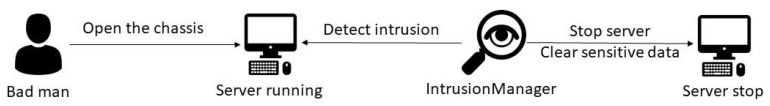
Diagram of the test case 1 process.

**Figure 23 sensors-22-01882-f023:**
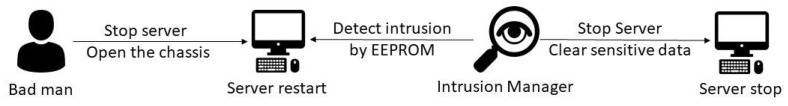
Diagram of the test case 2 process.

**Figure 24 sensors-22-01882-f024:**
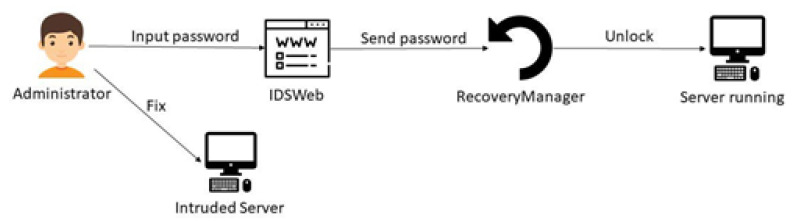
Diagram of the unlock process.

**Figure 25 sensors-22-01882-f025:**
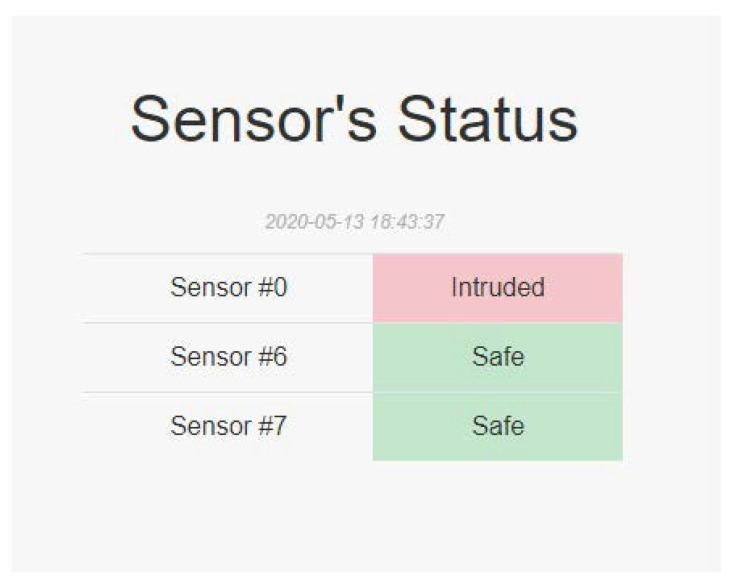
Sensor status.

**Figure 26 sensors-22-01882-f026:**
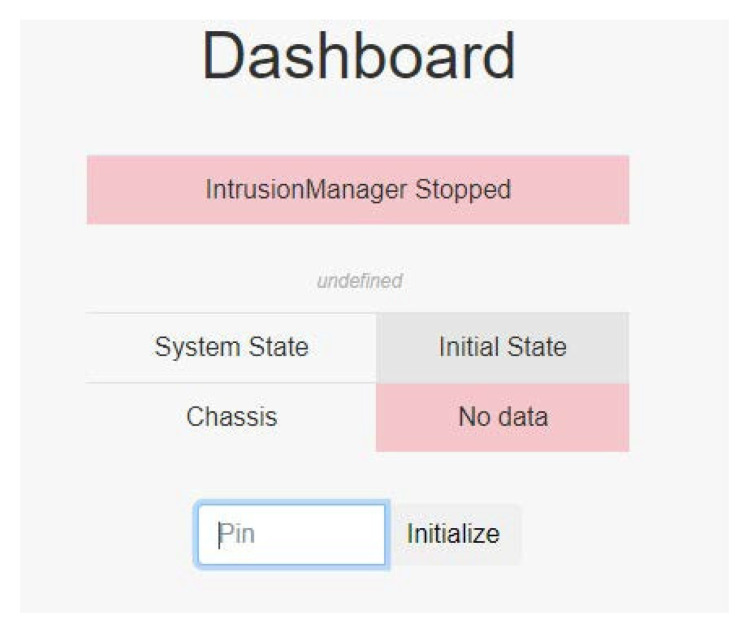
Initial state.

**Figure 27 sensors-22-01882-f027:**
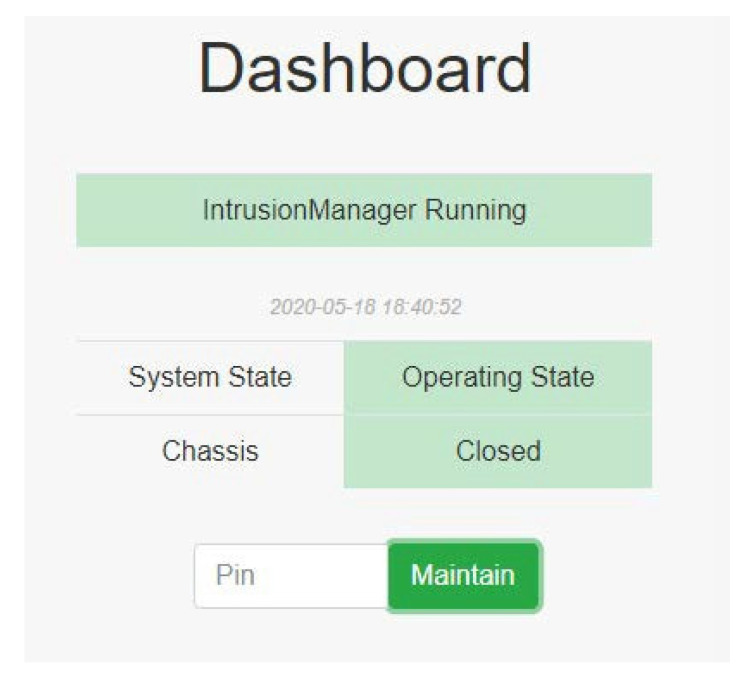
Operating state.

**Figure 28 sensors-22-01882-f028:**
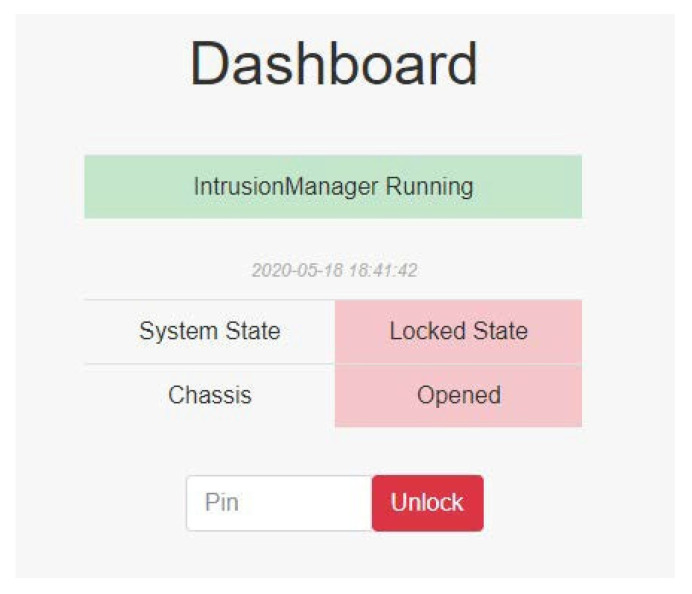
Locked state.

**Figure 29 sensors-22-01882-f029:**
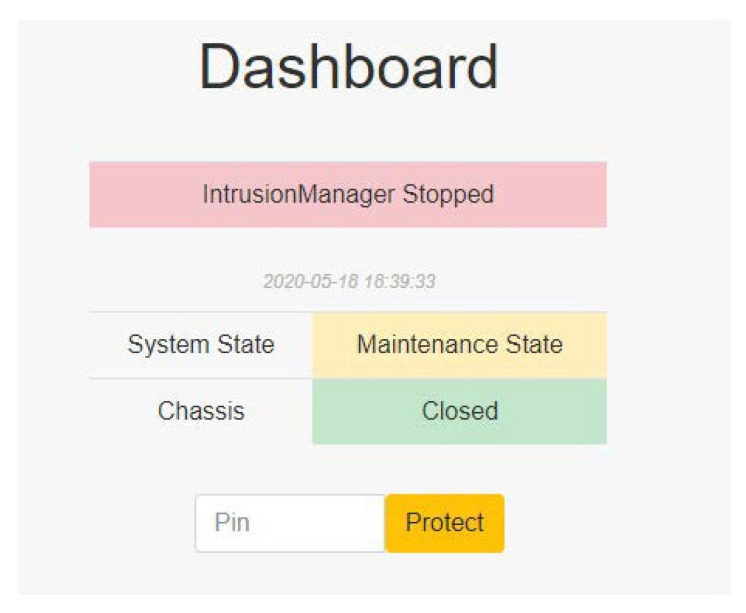
Maintenance state.

**Table 1 sensors-22-01882-t001:** Top five computer security vulnerabilities in 2019.

Rank	Vulnerabilities
1	Hidden Backdoor Programs
2	Superuser or Administration Account Privileges
3	Automated Running of Scripts without Malware/Virus Checks
4	Unknown Security Bugs in Software or Programming Interfaces
5	Unencrypted Data on the Network

**Table 2 sensors-22-01882-t002:** PSPS vs. HIDS-OSSEC vs. NIDS-SNORT.

IDS	PSPS	OSSEC	SNORT
Placement	Host	Host	Network
Monitoring	Computer chassis	System logs	Network packets
Alerts	Web/SMS/Email/Logs	Email/Logs	Logs
IPS	V	X	V

## Data Availability

Not applicable.

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
