# Peer review of "PSPS: A Step toward Tamper Resistance against Physical Computer Intrusion"

_sensors, 2022, doi:10.3390/s22051882_

Round 1
Reviewer 1 Report
- This paper needs a complete proof-of-read, the writing needs significant improvements. All the figures need to be polished, Figure 1 need citation.
- Lack of novelty
- The data of figure 1 and table 1 that is used for motivation of work is of 2019 while the paper is submitted at the end of 2020
- Abstract and conclusion need to be mapped with each other
- Web-link references need to be replaced with published work from peer review journals
Author Response
Dear Reviewer, The content of our paper has been rewritten, and the title has changed slightly. We make it more concise, Abstract and Conclusion are mapped with each other, and also have citation with Figure 1. We also add more experimental results in this contents.
One problem is that the citation from reference 1 updated to 2020 year, but the Total Malware Inflection Growth Rate still show until 2018 year, seeing the below:
reference 1 link: https://purplesec.us/resources/cyber-security-statistics/
Finally, we also update some recently year references.
Please see the attachment.
Thanks for your valuable suggestions.

Reviewer 2 Report
The paper needs a total improvement editing.
The manuscript is not ready for review and publication. The bibliography survey is poor, the research structure is not clear while the results are not presented or in details discussed.
The authors can redesign the paper and improve the quality.
Author Response
Dear Reviewer,
The content of our paper has been rewritten, and the title has changed slightly. We want to make it more concise, and the stated issue to be obvious, Abstract and Conclusion are mapped with each other, the test case contents of Implementation adding more descriptions and programing process in detail. Finally, we also update some recently year references.
Please see the attachment.
Thanks for your valuable suggestions.

Reviewer 3 Report
In this manuscript authors proposed a model for intrusion detection.
I cannot find any indication on the learning times and especially the time of recognition or validation of attacks. Since this recognition must be done in real time and in a very short period of time, the time constraint is a crucial parameter because the attack takes advantage of the weakness of the network during very short periods of time.
Please Add few more up to dated articles related to your problem in literature section.
An evaluation of the proposed solution in a real scenario is missing.
Simulation and result section not good. it needs major revision. Add more statistical analysis like sensitivity, specificity, TPR, and compared your model accuracy with previous well known published approaches like DELM, RTS-DELM etc.
Please mentioned the
MapReduce Based Intelligent Model for Intrusion Detection Using Machine Learning Technique. Journal of King Saud University-Computer and Information Sciences. 2021.
Author Response

(The authors gave the same response as above.)

Reviewer 4 Report
The topic is relevant. The introduction appears to be very concise, touching key concepts. The problem, the method, and the results are well described. The actual research hypothesis is not defined explicitly. "Our main purpose is to find out whether our system can discover the computer’s
chassis is opened," is the closest phrase, the stated issue is quite obvious, please explain where the problems lie. The test cases should be described in more detail: is the protection device hidden, or fast, or difficult to cheat: please explain what makes it unique and worth a publication.
In general: please have language checked (tenses, singular/plural, use of articles, punctuation marks)
Author Response

(The authors gave the same response as above.)

Round 2
Reviewer 2 Report
The revised version of the paper has been improved in many points.
However, the authors can improve the followings:
- Fig.13 needs copyright license or use your own photo
- Figs.15-18 can be write as text-box including the code in order to be clear.
- Conclusions chapter can be more extended (it is somehow short)
Author Response
Dear Reivewer, thank you for giving such valuable comments, please see the attachment.

Reviewer 3 Report
satisfied
Author Response
Dear Review, thank you for giving such some valuable comments. Please see the attachment.
